# Examining the Role of Traditional Masculinity and Depression in Men’s Risk for Contracting COVID-19

**DOI:** 10.3390/bs12030080

**Published:** 2022-03-16

**Authors:** Andreas Walther, Lukas Eggenberger, Jessica Grub, John S. Ogrodniczuk, Zac E. Seidler, Simon M. Rice, David Kealy, John L. Oliffe, Ulrike Ehlert

**Affiliations:** 1Department of Clinical Psychology and Psychotherapy, University of Zurich, 8050 Zurich, Switzerland; lukas.eggenberger@uzh.ch (L.E.); j.grub@psychologie.uzh.ch (J.G.); u.ehlert@psychologie.uzh.ch (U.E.); 2Department of Psychiatry, University of British Columbia, Vancouver, BC V6T 1Z3, Canada; john.ogrodniczuk@ubc.ca (J.S.O.); david.kealy@ubc.ca (D.K.); 3Centre for Youth Mental Health, The University of Melbourne, Parkville, VIC 3010, Australia; zac.seidler@movember.com (Z.E.S.); simon.rice@orygen.org.au (S.M.R.); 4Orygen, Melbourne, VIC 3052, Australia; 5School of Nursing, University of British Columbia, Vancouver, BC V6T 2B5, Canada; john.oliffe@ubc.ca; 6Department of Nursing, University of Melbourne, Parkville, VIC 3010, Australia

**Keywords:** traditional masculinity, traditional male role norms, COVID-19, depression, MDRS-22

## Abstract

In the light of the COVID-19 pandemic and claims that traditional masculinity may put some men at increased risk for infection, research reporting men’s health behaviors is critically important. Traditional masculine norms such as self-reliance and toughness are associated with a lower likelihood to vaccinate or follow safety restrictions. Furthermore, infection risk and traditional masculinity should be investigated in a differentiated manner including gender role orientation, underlying traditional masculine ideologies and male gender role conflict. In this pre-registered online survey conducted during March/April 2021 in German-speaking countries in Europe, 490 men completed questionnaires regarding contracting COVID-19 as confirmed by a validated test, fear of COVID-19 (FCV-19S), and experience of psychological burden due to COVID-19. In addition, depression symptomatology was assessed by using prototypical internalizing and male-typical externalizing depression symptoms. Furthermore, self-identified masculine gender orientation, endorsement of traditional masculinity ideologies, and gender role conflict were measured. A total of 6.9% of men (n = 34) reported having contracted COVID-19 since the beginning of the pandemic. Group comparisons revealed that men who had contracted COVID-19 exhibited higher overall traditional masculine ideology and gender role conflict. Logistic regression controlling for confounders (age, income, education, and sexual orientation) indicated that only depression symptoms are independently associated with the risk of having contracted COVID-19. While prototypical depression symptoms were negatively associated with the risk of having contracted COVID-19, male-typical externalizing depression symptoms were positively associated with the risk of contracting COVID-19. For traditional masculinity, no robust association for an increased risk of contracting COVID-19 could be established, while higher male-typical externalizing depression symptoms were associated with an increased risk of contracting COVID-19.

## 1. Introduction

The COVID-19 pandemic poses major health and economic ramifications worldwide and effectively responding to the pandemic remains an ongoing challenge. Investigation of the pandemic is critical to ending it and better controlling future pandemics [1]. As one such central concept that might underlie the increased spread of the COVID-19 virus, traditional masculinity is increasingly coming into focus [2].

Although men and women were infected with COVID-19 to a similar extent at the beginning of the pandemic, men were much more likely to die from COVID-19 [3]. Research indicates that the cause of the increased mortality rate in infected men was not rooted in male biology but in men’s health risk behaviors such as smoking and drinking, which can complicate COVID-19 infections and may lead to a more severe disease course [3,4]. Traditional masculinity in connection with health risk behavior has become increasingly important in explaining why specific sub-populations adhere less to state-imposed protective measures such as mask wearing or show increased mortality [2].

Traditional masculinity has long been related to men’s health risk behaviors [5,6]. This is reflected by consistent findings of gender differences emerging from the current pandemic showing that men were found to be more likely than women to downplay the risks associated with COVID-19 [7]. Men were also less likely than women to report fearing “very serious” consequences if they became infected [8,9]. Men in general were also less likely to follow public health protocols [10,11], including refusing to wear masks, demonstrating greater negative attitudes towards wearing a mask, and less reported handwashing and social distancing compared to women [9,11,12]. Relative to women, men are more likely to believe that wearing a face mask is embarrassing or a sign of weakness [9] and Palmer and Peterson [13] specifically linked stronger endorsement of the traditional male role norm of toughness with greater levels of negativity towards mask wearing. Taken together, men in general, show poorer adherence to protection measures and health behaviors and, therefore, are at increased risk for contracting COVID-19 in comparison to women [9,13,14,15].

These hesitant attitudes and behaviors to follow state-imposed protection measures for ending the COVID-19 pandemic by many men appear to be based on socially constructed and idealized male gender norms of strength, toughness, and self-reliance adopted during men’s gender role socialization [16,17]. Thus, studies comparing men with low and high adherence or conformity to traditional male role norms should provide further insight into the relation between traditional masculinity, following protection measures and the risk for contracting COVID-19. However, to date there are few available studies on this topic. A non-peer reviewed but publicly available report highlighted those men who identified as “completely masculine” were nearly three times more likely to report having been diagnosed with COVID-19 than men self-identifying as “mostly masculine” or “slightly masculine” [18]. Furthermore, the same authors reported that men self-identifying as “completely masculine” expressed less intention to get vaccinated for COVID-19 as men self-identifying as “mostly masculine” or “slightly masculine” [14]. Mahalik and colleagues [15] provided preliminary evidence that higher conformity to traditional masculine norms is associated with negative attitudes toward mask wearing during the ongoing pandemic and that conformity to traditional masculine norms interacts with more conservative political ideologies for the prediction of mask wearing attitudes.

The relationship between traditional masculine role norms and health behaviors or health outcomes in general is, however, complex. Greater adherence to traditional masculinity was initially associated with worse health behaviors or increased mortality [19]. A meta-analytic investigation revealed that conformity to traditional masculine norms is associated with overall worse mental health outcomes and reduced help-seeking [20]. Moreover, studies indicate traditional masculinity to be associated with delayed therapy initiation [21] and increased risk for suicide [22,23]. However, more recently, positive and health promoting aspects of traditional masculinity are being recognized [24]. The conceptualization and operationalization of masculinity has evolved over time, beginning in the gender role identity paradigm attributing gender-typical traits to men and women [25,26], followed by the gender role strain paradigm focusing on the adherence and conformity to traditional masculine role norms [27,28,29]. Based on the social constructionist understanding of masculinity, Levant and Wimer [30], for example, revealed that conforming to certain masculine norms acts as a protective buffer for some health behaviors, while others are consistently identified as risk factors. This was corroborated in a subsequent study showing that the traditional masculine role norm of primacy of work was mostly related with positive health outcomes, while for four other role norms (winning, risk-taking, pursuit of status, and disdain for homosexuals) a balance of positive and negative health outcomes was observed [24]. Yet, six subscales (emotional control, violence, power over women, dominance, playboy, and self-reliance) were predominantly related to negative health outcomes [24]. Therefore, besides reliance on total scores of commonly used masculinity measures, which often obscure the complexity of associations, subscales highlighting particular male role norms should be investigated and reported.

In addition, it is increasingly being discussed whether individuals with mental disorders might be at increased risk for contracting COVID-19 due to potential cognitive deficits including executive dysfunction, negative health behaviors (e.g., smoking), or structural barriers hindering the ability to successfully quarantine at home [31,32]. While there are reports suggesting depression is associated with an increased risk for contracting COVID-19 [33,34,35,36], a similar number of studies found no association or a reduced risk for contracting COVID-19 in these individuals [37]. There is also evidence that some individuals intentionally expose themselves to the virus with the intent of self-harm or to suicide, further suggesting increased risk for contracting COVID-19 among individuals with mental health problems [38]. However, a meta-analysis examining the relationship between mood disorders and the risk of contracting COVID-19 in over 91 million individuals could not identify significant associations, suggesting the need to examine more specific subgroups regarding this question [39].

As no association was identified in the above-mentioned meta-analysis on the relation between mood disorders and the risk of contracting COVID-19 [39], it emerges as relevant to investigate a more fine-grained research question regarding how certain disorders, and indeed their unique symptomatology, are associated with a risk of contracting COVID-19. Considering that individuals with prototypical depression symptoms such as depressive mood, anhedonia, or fatigue show increased social withdrawal, higher rates of social isolation and unemployment, one would assume a reduced risk of contracting COVID-19 due to the reduced social interactions [39]. In contrast, men with high traditional masculinity exhibit more male-typical externalizing depression symptomatology [21,40], which is characterized by anger and aggression, risk-taking, or substance misuse [40,41,42,43,44,45]. Such a symptom pattern might be assumed to be associated with an increased risk of contracting COVID-19. Risk-taking, for example, is characterized by not caring about the consequences of one’s actions, so men experiencing externalizing depression symptomatology may care less about their own well-being or the well-being of those around them.

It is, therefore, important to emphasize that the constructs of traditional masculinity and depression are not independent, as the endorsement of traditional masculinity ideologies directly influences the presentation of depressive symptoms [40,41,42,43,44,45]. While men with low endorsement of traditional masculinity ideologies and mental distress are more likely to exhibit prototypical depression symptoms, men with high endorsement of traditional masculinity ideologies and mental distress are more likely to show male-typical externalizing depression symptoms [40,41,42,43,44,45]. Thus, the association between the risk of contracting COVID-19 and traditional masculinity is not independent of depression symptomatology presentation and could possibly be represented by it rather than by a direct link between the endorsement of traditional masculinity ideologies and COVID-19 infection risk.

The aim of the present study was to investigate the relationship between traditional masculinity measured by different conceptual constructs (gender role orientation, traditional masculine ideology, gender role conflict), depressive symptomatology (prototypical and male-typical externalizing depression symptoms), and self-reported infection with COVID-19. Based on the outlined literature, we hypothesize traditional masculinity to be positively associated with having contracted COVID-19. We hypothesize this relationship uniformly for all masculinity constructs and their respective subscales. The a priori formulated directed hypotheses are retrievable under OSF (https://osf.io/q4pw3, accessed on 9 March 2022). To better understand the relationship between specific depression symptomatology and the risk of contracting COVID-19, we further investigated, in an explorative manner, whether men with higher prototypical depression symptoms are less likely to contract COVID-19 and whether men with higher male-typical externalizing depression symptoms are more likely to contract COVID-19.

## 2. Materials and Methods

### 2.1. Sample and Procedure

This anonymous, cross-sectional online survey entitled ‘Men’s Mental Health in Times of COVID-19′ was pre-registered and approved by the local ethics committee of the Faculty of Arts and Social Sciences of the University of Zurich (Authorization No. 21.2.4). Following the Open Science standards, a priori defined study hypotheses, statistical analyses, and the study specific data set can be retrieved from OSF (https://osf.io/q4pw3, accessed on 9 March 2022). For this study, male participants were recruited via advertisements distributed on social media platforms such as Facebook and the study’s webpage. Advertisements on social media platforms were restricted to men of 18 years or older in the countries of Germany, Switzerland, Austria, Lichtenstein, Luxembourg, and Belgium. Aiming to recruit a large sample of men older than 18 years with sufficient German language skills to read and respond to the questionnaire in German language, all men irrespective of mental health status were eligible to participate. During the recruitment period from 15 March 2021 to 28 April 2021, a total number of 1087 people expressed interest in this study by visiting the starting page of the online questionnaire. A little more than half of the initially interested participants (n = 597, 54.92%) were not included in the final analyses for one of the following reasons: data privacy agreement not provided, declaration of consent not provided, self-reported insufficient German language skills, gender requirements not fulfilled, age of minority and/or incomplete data in the questionnaires. This resulted in a total number of 490 participants included in the analyses. Figure 1 presents participant flow. After providing written informed consent at the beginning of the survey, participants agreed to the data privacy statement and then went on to answer sociodemographic questions, COVID-19-related questions, and a set of psychometric instruments. For the subsequently described study, the average completion time was 20 min. However, several participants went on to participate in further online experiments following the completion of initial questionnaires, which are beyond the scope of this report. Further information on the study is provided elsewhere [22] or on the preregistration in OSF: “osf-link”.

### 2.2. Instruments

#### 2.2.1. Sociodemographics and COVID-19 Related Questions

The online survey started with sociodemographic questions assessing the sufficiency of German skills, gender, dimensional gender (positioning between the two poles 1 = masculine, 10 = feminine), age, height, weight, nationality, relationship status, sexual orientation, education, and the household’s yearly gross income. Participants were further asked whether a validated test confirmed that they were currently infected with COVID-19 or whether a validated test has previously confirmed that they had contracted COVID-19 since the beginning of the pandemic (yes/no). If participants answered “yes” to this question, they were asked which symptoms from a given list they currently or previously suffered from due to COVID-19 infection (e.g., respiratory symptoms, cardiovascular symptoms, fatigue, fever, pain symptoms, post-exertional malaise, cognitive symptoms, nausea, etc.). The list of symptoms was based on previous research investigating COVID-19 and long COVID symptoms [46]. Further descriptives are presented in Table 1.

#### 2.2.2. Male Role Norm Inventory–Short Form (MRNI–SF)

The Male Role Norms Inventory–Short Form (MRNI–SF) consists of 21 items and measures of traditional masculine ideology with seven subscales [47]. The participant indicates the degree to which he agrees with traditional masculinity ideology statements on a seven-point Likert scale (1 = strongly disagree, 7 = strongly agree). The seven subscales of the MRNI–SF represent the following dimensions: restrictive emotionality, self-reliance through mechanical skills, negativity towards sexual minorities, avoidance of femininity, importance of sex, dominance, toughness. A Cronbach’s alpha of α = 0.92 for men has been reported for the original English version of the MRNI–SF [47]. Cronbach’s α in the current study: MRNI–SF total score = 0.94, MRNI–SF negativity towards sexual minorities = 0.87, MRNI–SF restrictive emotionality = 0.74, MRNI–SF self-reliance through mechanical skills = 0.88, MRNI–SF avoidance of femininity = 0.88, MRNI–SF importance of sex = 0.88, MRNI–SF dominance = 0.88, MRNI–SF toughness = 0.81. 

#### 2.2.3. Gender Role Conflict Scale–Short Form (GRCS–SF)

The Gender Role Conflict Scale [29] measures patterns of gender role conflict (GRC). According to O’Neil, GRC describes a psychological state in which socialized gender roles have negative consequences for the person or others and occurs when rigid, sexist, or restrictive gender roles cause restriction, devaluation, or violation of others or self. The Gender Role Conflict Scale–Short Form [48] measures the following four patterns of gender role conflict: (1) success, power, and competition (SPC), (2) restrictive emotionality (RE), (3) restrictive affectionate behavior between men (RABBM), and (4) conflict between work and family relations (CBWFR). Consisting of a total number of 16 items, four items are dedicated to each pattern of GRC. The participants indicate the degree of experienced conflict on a six-point Likert scale (0 = strongly disagree, 5 = strongly agree). Wester and colleagues [48] reported internal consistencies of α = 0.77–0.80. The German version of the Gender Role Conflict Scale–Short Form was used in the present study [49]. Cronbach’s α in the current study was at 0.79.

#### 2.2.4. Patient Health Questionnaire-9 (PHQ-9)

The Patient Health Questionnaire (PHQ-9) [50] assesses nine symptoms of major depressive disorder specified by the Diagnostic and Statistical Manual of Mental Disorders 5 (DSM-5) [51]. For each of the nine symptoms, the participants rate how often they appeared within the preceding two-week period on a four-point Likert scale ranging from 0 (not at all) to 3 (almost every day). The PHQ-9 is applied in criteria-based diagnoses of depressive disorders with a cut-off ≥10 within research and clinical practice [52] and has been shown to provide a reliable and valid measure of depression severity [50]. In this study, a German version of the PHQ-9 was used, which has been previously validated on a representative German-speaking sample (Cronbach’s α = 0.89) [53]. Cronbach’s α in the current study was at 0.90.

#### 2.2.5. Male Depression Risk Scale-22 (MDRS-22)

The Male Depression Risk Scale (MDRS-22) consists of 22 items assessing externalized depressive symptoms [40]. For each item, the participant rates how often the symptom appeared within the preceding month on an eight-point Likert scale ranging from 0 (not at all) to 7 (almost always). The scale consists of six domains and considers gender-specific externalizing symptoms of depression and assesses them within the context of cultural norms related to masculinity. The six domains of the MDRS-22 include emotion suppression (i.e., “I bottled up my negative feelings”), drug use (i.e., “I used drugs to cope”), alcohol use (i.e., “I needed to have easy access to alcohol”), anger and aggression (i.e., “I overreacted to situations with aggressive behaviors”), somatic symptoms (i.e., “I had regular headaches”). In the present study, the validated German version of MDRS-22 was used (Guttman’s λ_2_ = 0.62–0.91) [43]. Cronbach’s α in the current study was at 0.88.

#### 2.2.6. Fear of COVID-19 Scale (FCV-19S) and COVID-19 Pandemic Stress Scale (CPSS)

As a measure for the evaluation of discriminate validity, with regard to depression measures, we included the German version of the fear of COVID-19 scale [54], consisting of 7 items where participants have to indicate the extent to which they do agree with the statements on fear of COVID-19 on a five-point Likert scale (1 = do not agree at all; 5 = fully agree). Cronbach’s α in the current study was at 0.85. We further included the German version of the COVID-19 Pandemic Stress Scale, consisting of 10 items on aspects and consequences of COVID-19, such as hygienic behavior rules, contact restrictions, or actual fear of COVID-19 infection [55]. The participants had to rate how stressed or anxious they felt about these topics during the previous two weeks on a four-point Likert scale (0 = not burdened at all, 4 = very burdened). Cronbach’s α in the current study was at 0.74.

### 2.3. Statistical Analysis

The statistical analyses and computations were performed with the software R [56] and the supplementary packages “psych” [57] for calculating internal consistencies, effect sizes, and correlations, “car” [58] for estimating variance inflation factors, “rcompanion” [59] for MLE estimation of goodness-of-fit indices for logistic regression models, and “ggplot2” [60] for data visualization. The subsamples used for the analyses described in the following were obtained by dividing the participants into two groups consisting of men who (previously or currently) tested positive for COVID-19 and men who never tested positive for COVID-19. While for the initial analyses, a significance level of α = 0.05 was used, a sequential Holm–Bonferroni correction for multiple testing was applied post-hoc to control the family-wise error rate in each separate step of the analysis [61]. Additionally, due to the a priori formulated hypothesis, one-sided hypothesis tests were used in all the analyses involving traditional masculinity constructs (BSRI, MRNI–SF, GRCS–SF, including all their subscales). Lastly, statistical assumptions were assessed using the Levene’s test for the homogeneity of variance [62] for the *t*-tests and the generalized variance inflation factor [63] as well as Cook’s distance [64] for the logistic regression models. Because the distribution of the MDRS-22 and MRNI–SF scores was quite skewed in some intervals (Appendix A), the *p*-values of the t-tests from these two variables, including the subscales of the MRNI–SF, were obtained by bootstrapping with 5000 repetitions.

Initially, sample characteristics were obtained by calculating mean scores and frequency distributions for the total sample as well as for the two subgroups separately. Additional *t*- and χ^2^-tests were then used to test for statistically significant group differences between the two subgroups. Secondly, correlational analyses were conducted by calculating the Pearson’s correlation coefficient for the relevant variables and subsequently testing their significance with two-sided *t*-tests. Thirdly, logistic regression analyses were performed to assess the predictive value of depressive symptoms, traditional masculinity, fear of COVID-19, and COVID-19 pandemic-related stress. These regression models included the participant’s age, income, education level, and sexual orientation as covariates. In a last part, a more explorative approach was used to assess a possible association between traditional masculinity and COVID-19 symptoms. For this purpose, men who have contracted COVID-19 were divided into two subgroups using median division of the variables assessing the traditional masculinity constructs so that men with high and low traditional masculinity could be identified and compared with regard to displaying specific COVID-19 symptoms. Wald-tests were then used to determine significant differences in the frequency of COVID-19 symptoms and *t*-tests were used to determine significant differences in the mean scores of the remaining questionnaires used in the study.

## 3. Results

### 3.1. Descriptive Statistics and Group Differences in Men with and without COVID-19 Infection

Out of the 490 men participating in the survey, 34 (6.9%) previously or currently tested positive for COVID-19, which corresponded to the COVID-19 positive rates at that time for the surveyed countries Germany (4.1%), Austria (6.9%), and Switzerland (1.5%). Their age ranged from 18 to 68 years old, with a mean age of 25.7 years. The majority of the participants originated from Germany (73.1%) and Switzerland (14.5%), self-identified as heterosexual (73.7%), were single (63.5%), and had completed secondary education (71.0%). More detailed sample demographics can be found in Table 1.

Regarding group differences (Figure 2, Appendix A), men who had contracted COVID-19 had significantly higher MRNI–SF scores on the total scale (Cohen’s d = 0.44) as well as on the MRNI–SF subscales assessing Restrictive Emotionality (RE; d = 0.38), Importance of Sex (IS; d = 0.31), Dominance (D; d = 0.55), and Toughness (T; d = 0.30). Similarly, men who had contracted COVID-19 also had significantly higher GRCS scores on the total scale (d = 0.40) as well as on the subscale measuring Success, Power, Competition (SPC; d = 0.38). However, none of these effects remained significant after applying the Holm–Bonferroni correction for multiple testing to all inferences made in this step.

### 3.2. Correlational Analysis

As presented in Table 2, the PHQ-9 was significantly positively correlated with the MDRS-22 (*r* = 0.64), the GRCS (*r* = 0.41), the FCV-19S (*r* = 0.29), and the CPSS (*r* = 0.31), while also being significantly negatively correlated with the BSRI (*r* = −0.36). The MDRS-22 was significantly positively correlated with the MRNI–SF (*r* = 0.18), the GRCS (*r* = 0.38), the FCV-19 (*r* = 0.28), and the CPSS (*r* = 0.28). The BSRI was only significantly positively correlated with the MRNI–SF (*r* = 0.33) while also being significantly negatively correlated with the FCV-19S (*r* = −0.16). The MRNI–SF was again significantly positively correlated with the GRCS (*r* = 0.39) and significantly negatively correlated with the FCV-19S (*r* = −0.15). The GRCS was significantly positively correlated with the FCV-19S (*r* = 0.18) and the CPSS (*r* = 0.24). Lastly, the FCV-19S was significantly positively correlated with the CPSS (*r* = 0.58). All effects in this part of the analysis remained significant after applying the Holm–Bonferroni correction. A more detailed overview, including the various sub-scales, is provided in the Appendix A. 

### 3.3. Logistic Regression Predicting COVID-19 Infection

For the binary logistic regression analysis, two models were fitted due to singularities in the covariance matrices caused by perfect linear combination among the MRNI–SF total score and its subscales, as well as the GRCS–SF and its subscales. The first model therefore included the total scores of the MRNI–SF and the GRCS–SF as predictors for a COVID-19 infection (Appendix A) and revealed the PHQ-9 (OR = 0.91 (0.83–0.99), standardized OR = 0.53 (0.30–0.93)) and the MDRS-22 (OR = 1.03 (1.00–1.05), OR_std._ = 1.66 (1.04–2.65)) to be the only two significant predictors for a COVID-19 infection. The second model (Figure 3) included only the subscales of the MRNI–SF and the GRCS–SF instead of the total scores, but the same two predictors still emerged as significant (PHQ-9: OR = 0.89 (0.81–0.98), OR_std._ = 0.47 (0.25–0.85); MDRS-22: OR = 1.02 (1.00–1.05), OR_std._ = 1.64 (1.01–2.66)). Both models, therefore, indicate higher PHQ-9 scores to be associated with a lower likelihood of contracting COVID-19 and higher MDRS-22 scores to be associated with a higher likelihood of contracting COVID-19. However, these results also became non-significant after applying the Holm-Bonferroni correction for multiple testing to all inferences made in this step simultaneously. Nonetheless, further post-hoc analyses favored the first model with the total scores of the MRNI–SF and the GRCS–SF (BIC = 297.7, χ^2^(11) = 23.7, *p* = 0.015) as a better overall fit in predicting a COVID-19 infection as compared to the second model including only the subscales of the MRNI–SF and the GRCS–SF (BIC = 349.3, χ^2^(20) = 27.8, *p* = 0.114).

### 3.4. Group Differences in Men with Low and High Traditional Masculinity

An explorative analysis was taken to further examine possible associations between high traditional masculinity, COVID-19 symptoms, and depressive symptomatology in men who contracted COVID-19. Here, only the subsample consisting of 34 men was used. As can be seen in Figure 4A, men with high traditional masculinity operationalized as high BSRI scores exhibited less general (i.e., pain symptoms, headache, joint pain, muscle pain, peripheral neuropathy) pain symptomatology (47.4%) as compared to men with low traditional masculinity (80.0%) (χ^2^(1) = 4.22, *p* = 0.040, OR_std._ = 0.17 (0.03–0.84)) and lower PHQ-9 scores (*M* = 9.2, *SD* = 5.7) as compared to men with low traditional masculinity (*M* = 13.9, *SD* = 6.3) (*t*(32) = −2.26, *p* = 0.031, *d* = 0.37). Furthermore, as can be seen in Figure 4C, men who experienced higher gender role conflict exhibited higher MDRS-22 scores (*M* = 44.5, *SD* = 21.5) as compared to men who experienced lower gender role conflict (*M* = 23.8, *SD* = 23.1) (*t*(32) = 2.70, *p* = 0.011, *d* = 0.43). However, these results also faded after applying the correction for multiple testing to all inferences made in this step simultaneously.

## 4. Discussion

### 4.1. Summary of Results

The present study examined the relationships between mental health outcomes, traditional masculinity, and COVID-19 infection. The first set of hypotheses postulated that higher expression of traditional masculinity constructs (gender role orientation, traditional masculine ideologies, gender role conflict) would be associated with a higher likelihood of having contracted COVID-19. Although group comparisons between men infected with COVID-19 and those without infection showed that infected men exhibit significantly elevated scores of traditional masculine ideologies and gender role conflict (but not gender role orientation), this could not be confirmed in the logistic regression models controlling for the confounders of age, income, education, and sexual orientation. Notably, when applying correction for multiple testing, the significant group differences did not hold.

In a second step, the hypotheses that men with elevated prototypical depression symptoms would have a lower risk of being infected with COVID-19, whereas men with elevated male-typical externalized depression symptoms would have an increased likelihood of being infected with COVID-19 were tested. Although this tendency was observable in the group comparisons, no significant group differences emerged. However, these hypotheses were subsequently confirmed in the logistic regression analyses controlling for confounders. It is important to note that when the Holm–Bonferroni correction for multiple testing was applied, the significant effects did not hold.

### 4.2. Integration of Findings

The results partly support our assumption that higher expression of traditional masculinity (MRNI–SF and GRCS–SF but not BSRI-M) is associated with a higher likelihood of having contracted COVID-19. Two out of the three masculinity measures were significantly elevated in the group of men reporting having contracted COVID-19 since the beginning of the pandemic as compared to men without known COVID-19 infection. Specifically, for the endorsement of traditional masculinity ideologies (MRNI–SF), the subscales of restrictive emotionality, importance of sex, dominance, and toughness were elevated in the men with a positive COVID-19 test, while for gender role conflict (GRCS–SF), the subscale of power, competition, and success was elevated. Though, only reaching statistical significance in the univariate analysis, an increase of the MRNI–SF total score by one standard deviation suggests an increase in the likelihood for a COVID-19 infection by 1.12 (OR). As shown in Appendix A, the MRNI–SF subscale Dominance (OR = 1.46) shows one of the strongest associations with COVID-19 infection risk of all traditional masculinity scores. A man having a higher score in the dimension of Dominance by one standard deviation would, therefore, be 1.46-times more likely to have contracted COVID-19 than a man with a lower score. These findings are consistent with previous studies showing that adherence to traditional masculine norms is associated with engagement in risky health behaviors in men [30,65]. The present study thereby partly corroborates the previous literature on traditional masculinity and health behaviors in general, but also with regard to mask wearing during the current pandemic [15], extending it with regard to the most critical outcome to fight the current Corona pandemic, namely contracting COVID-19.

Importantly, a non-peer reviewed report examining over 6000 adults suggested that men self-identifying as “completely masculine” were almost three times more likely to contract COVID-19 than men self-identifying as “mostly masculine” or “slightly masculine” [18]. Data stemming from the present study could not confirm such a direct association for any of the three validated masculinity measures, highlighting three potential explanations for the observed lack of direct association. Firstly, it could be that included confounders (age, education, income, sexual orientation) better explain COVID-19 infection risk than traditional masculinity. Secondly, the study might have been unable to detect a signal due to the number of positive COVID-19 infection cases. Thirdly, it is imaginable that using validated masculinity scales reveals a more complex picture than reported by Cassino [18] using a single item to measure masculinity. This is in line with a previous report suggesting that total scores obscure the complex relationship between several traditional masculine norms and health behaviors or outcomes [24]. This study further highlighted, that in using the Conformity to Masculine Norms Inventory, with its subscales as a measure for conformity to traditional masculinity, 30% of the findings reflected beneficial associations with specific masculine role norms and health or well-being outcomes, especially for health promotion. Therefore, the observed findings of uniformly increased traditional masculinity in the group having contracted COVID-19 (see Figure 2) are pointing towards a relationship between traditional masculinity ideology and COVID-19 infection risk, although the Holm–Bonferroni correction renders the effects null. A further study from our workgroup is currently underway with the goal of replicating the observed findings with the advantage of accessing a larger population of contracted individuals with less time restrictions for recruitment due to the largely stabilized pandemic situation.

It is interesting that when predictors of traditional masculinity (MRNI–SF, GRCS–SF) are included separately in a univariate model along with covariates, they result as significant predictors of the risk of contracting COVID-19 as shown in Appendix A, whereas when they are combined with other relevant predictors such as fear of COVID-19 or depressive symptomatology in multivariate analyses, they no longer become significant (see Appendix A). On the one hand, it seems that traditional masculinity measures explain the risk of contracting COVID-19 through the dimensions of “dominance”, “restrictive emotionality”, or “toughness”, which stand opposed to fear of COVID-19. In line with this, men with higher traditional masculinity also report less fear of COVID-19 and its consequences [8,9]. However, since the variance of contracting COVID-19 is co-explained in multivariate analyses by the Fear of COVID Scale, this arguably leads to these constructs subtracting mutually explained variance from each other resulting in null associations. Similarly, depression symptoms may co-explain the risk of contracting COVID-19 in multivariate analyses. Since, in this study, the measure of endorsement of traditional masculinity ideologies (MRNI–SF) is negatively associated with prototypical depression symptomatology (PHQ-9) on the one hand, but positively associated with male-typical externalized depression symptomatology (MDRS-22) on the other (see Table 2 and Appendix A for subscale relations), a partitioning of the variance explanation between the endorsement of traditional masculinity and depression symptoms is likely to occur as well. Here, it appears that both forms of depression symptomatology combined with the constructs of traditional masculinity and COVID-19-related anxiety and burden are the strongest predictors of the risk of becoming infected with COVID-19 in the multivariate analyses. Nevertheless, this explanation must be considered tentative and tested in larger samples with more statistical power. In such larger samples, there may be sufficient statistical power to identify traditional masculinity measures as robust significant predictors.

The fact that men with high traditional masculinity exhibit worse health behaviors including more risk-taking and less mask-wearing [15] and, thus, are exposed to a higher risk of infection, also further explains why men are more likely to die from COVID-19 in comparison to women [3,4]. The present study sheds light on why men, and particularly men with high traditional masculinity, may be at increased risk for contracting COVID-19 and, thus, potentially increased mortality as shown in previous longitudinal studies [19,23].

Furthermore, results supported the exploratory hypotheses that higher prototypical depression symptoms (PHQ-9) are associated with a reduced risk of contracting COVID-19, while higher male-typical externalized depression symptoms (MDRS-22) are associated with an increased risk of contracting COVID-19. Thus, an increase in the PHQ-9 total score by one standard deviation lowers the likelihood for a COVID-19 infection by 0.53 (OR), while an increase in the MDRS-22 total score by one standard deviation increases the likelihood for a COVID-19 infection by 1.66 (OR). This can be interpreted as a man with a total PHQ-9 score 6.6 points higher than another man is only nearly half as likely to contract COVID-19, while a man with a total MDRS-22 score 20.6 points higher than another man is 1.66-times more likely to contract COVID-19. This finding is new insofar that no previous study has examined prototypical and male-typical externalized depression symptoms in parallel with regard to the risk of contracting COVID-19. The observation that prototypical depression symptoms, likely due to their relationship with a general social withdrawal, are associated with a reduced infection risk, has previously been reported by independent studies [39]. However, the contrary or a null association has also been reported several times [33,34,35,36], suggesting that this relationship is either non-existent or depends on examined subgroups and the specific depression symptomatology exhibited. A previous study from our workgroup examining adult men’s psychotherapy use found that only higher male-typical externalized depression symptoms, but not prototypical depression symptoms, predicted lower psychotherapy use [21]. Similarly, male-typical externalizing and prototypical depression symptoms were shown to differ in predicting a past-month suicide attempt [22,66]. These findings support the notion that internalizing and externalizing psychopathology are differentially related to health behavior risk and specifically to the risk of contracting COVID-19. However, it is also relevant that COVID-19 increased the risk of suffering from depression measured with the PHQ-9 more than 3-fold [67], which additionally highlights the importance to understand which individuals suffer from which depression symptoms and whether they are at particular risk of contracting COVID-19.

In summary, the present data support the hypothesis that traditional masculinity and the presence of male-typical externalized depression symptoms may be associated with an increased risk of contracting COVID-19. Therefore, health promotion campaigns should specifically target these men, as they might be relatively poorly informed due to a more indifferent attitude based in traditional masculinity ideologies of dominance, toughness, and self-reliance, or in extreme cases these men may also deliberately infect themselves with COVID-19 to self-harm or suicide [38]. One way to address this problem would be to appeal to the roles of protector and provider and ask men with high traditional masculinity to protect others and not put them in danger, which also means protecting themselves so as not to be carriers of the virus.

Mental health care specialists should pay particular attention to men with high traditional masculinity and male-typical externalizing depression symptoms and inform them about the detrimental personal and societal consequences a careless spreading of the virus has. The effect men with high traditional masculinity and male-typical externalizing depression symptoms have in spreading the virus, and thereby prolonging the current pandemic situation, is difficult to identify. Nevertheless, informing men and the entire health care system about the crucial role men with high traditional masculinity may play in this pandemic is of vital importance. However, the use of a strength-based approach by appealing to the roles of protector and provider or good leader may be all the more promising in limiting the risk of COVID-19 infection by men with high traditional masculinity as a means of protecting significant others.

It is important to note, however, that the present findings can only be considered preliminary results since limited power caused significant results to fade after correction for multiple testing. Due to a time-sensitive measurement period during the COVID-19 pandemic, it was not possible to recruit participants for a sufficiently long period and, therefore, only a small sample of 490 men in total, and 34 currently or previously COVID-19-infected men, could be obtained. Therefore, the present findings based on the a priori formulated hypotheses are to be considered tentative. The observation that findings uniformly fade when applying correction for multiple testing is, however, also to be interpreted with caution since no prior results on this specific research question is available and potentially a substantially larger sample size would be required to uphold the small to moderate findings in the range of statistical significance due to the small number of positive COVID-19 cases. Due to the time restrictions set by a dynamic pandemic situation, this poses a major challenge for research and, thus, corrections for multiple testing must be placed in perspective.

Of further interest are the observations regarding traditional masculinity and specific COVID-19-related symptoms (see Figure 4). A detailed discussion of these findings is provided in the Appendix A.

### 4.3. Limitations

When interpreting the study results, some limitations should be taken into account. First, the cross-sectional design only provides information about associations and no causal conclusions can be drawn from the results. Furthermore, the sample with a total of 490 men and 34 men with a reported COVID-19 infection is not sufficiently powered to identify small effects. Therefore, all significant results faded after the Holm–Bonferroni correction. Furthermore, the main outcome measure of a positive corona infection by a validated test has been self-reported, and because of insufficient data, little can be said about its reliability despite its high face validity. It also becomes evident from the descriptive data showing a high percentage of men suffering from a psychological disorder or suicidal thoughts, that this is a sample of adult men with high psychological distress, which makes it difficult to generalize to the male population as a whole. Further, it should be noted that the PHQ-9 and MDRS-22 depression scales, while very well-validated measures of depression, are not capable of identifying depressive disorders and, thus, interpretations regarding depression status and risk for contracting COVID-19 in men need to be made on a syndrome level, while a clinical diagnostic interview enabling differential diagnosis would be preferable. Another limitation of the study may be the analysis of the relationships with logistic regressions as it was pre-specified in the preregistration process. In future investigations, structural equation modeling approaches could be used to better represent the complex network of constructs involved. Thus, it could be directly examined whether associations are mediated by third variables. Furthermore, an investigation in a longitudinal design could examine the covariation of depression symptoms and traditional masculinity with respect to their influence on relevant health outcomes, such as COVID-19 infection. Lastly, the fact that there is partial overlap between the subscales of the instruments complicates the analytic strategy and interpretation of the results. However, the aim of the study was to compare the different masculinity instruments, which are based on three different conceptual derivations, with respect to the risk of contracting COVID-19. Thus, the BSRI-M is derived from the Gender Role Identity Paradigm and describes self-attributed masculine traits, whereas the MRNI–SF is derived from the Gender Role Strain Paradigm and describes the endorsement of traditional masculinity ideologies. The GRCS–SF, on the other hand, measures the rigid adherence to or deviation from traditional male role norms and its negative psychological consequences for the man or his environment. Therefore, these constructs are each distinct from one another and can independently elucidate variance for the pre-specified research questions. Due to the fact that there was no previous research in this area, it emerged as important to test different relevant instruments associated with traditional masculinity with regard to the risk of contracting COVID-19. This was in the pursuit of a comprehensive investigation without previous research findings on the specific topic, while at the same time leading to a co-explanation of the primary outcome by these constructs. Therefore, we provide the interested reader with multivariate and univariate analyses at the total score and subscale levels in Appendix A, as well as the variance inflation factors testing for multi-collinearity showing no multi-collinearity for the used constructs in Appendix A.

## 5. Conclusions

Taken together, men exhibiting higher traditional masculinity and higher male-typical externalized depression symptoms may be at increased risk for contracting COVID-19 and should be informed by health care campaigns and mental health care specialists. Appealing to these men’s provider and protector roles might partially counter the overall detrimental effects of high traditional masculinity without challenging the endorsement of traditional masculinity ideology itself. The potential effect men with high traditional masculinity and increased male-typical externalized depression symptoms have in prolonging the current pandemic is difficult to identify. More research using validated traditional masculinity measures, combined with larger samples including more men with current or past COVID-19 infections, is needed.

## Figures and Tables

**Figure 1 behavsci-12-00080-f001:**
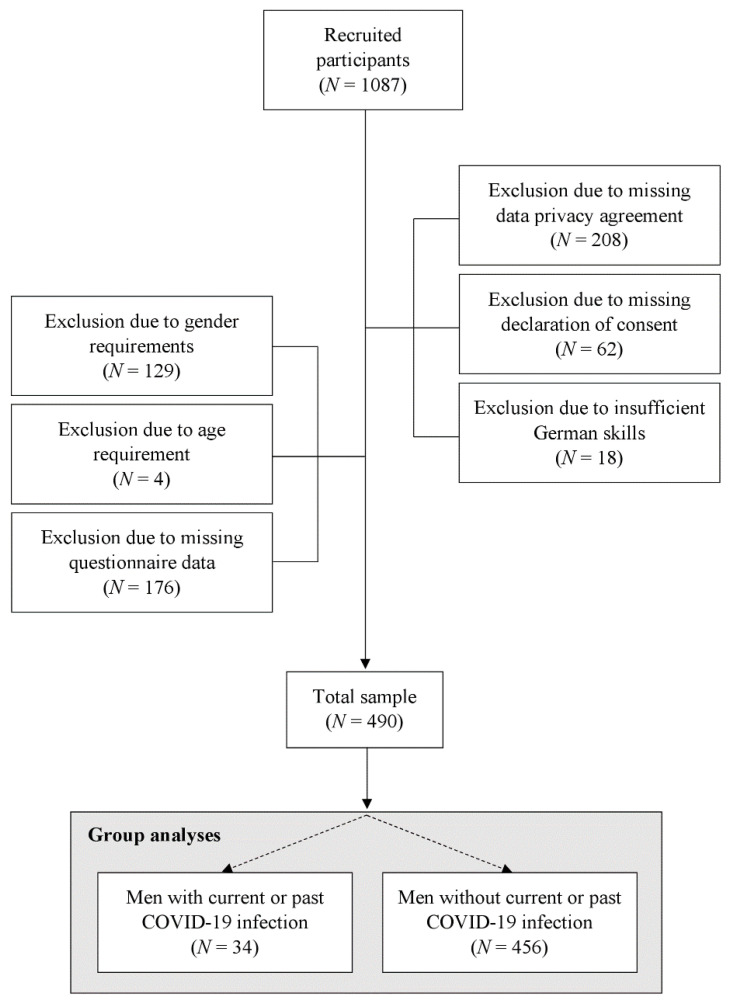
Flow diagram of the inclusion and exclusion process. Note: *N* = number of participants.

**Figure 2 behavsci-12-00080-f002:**
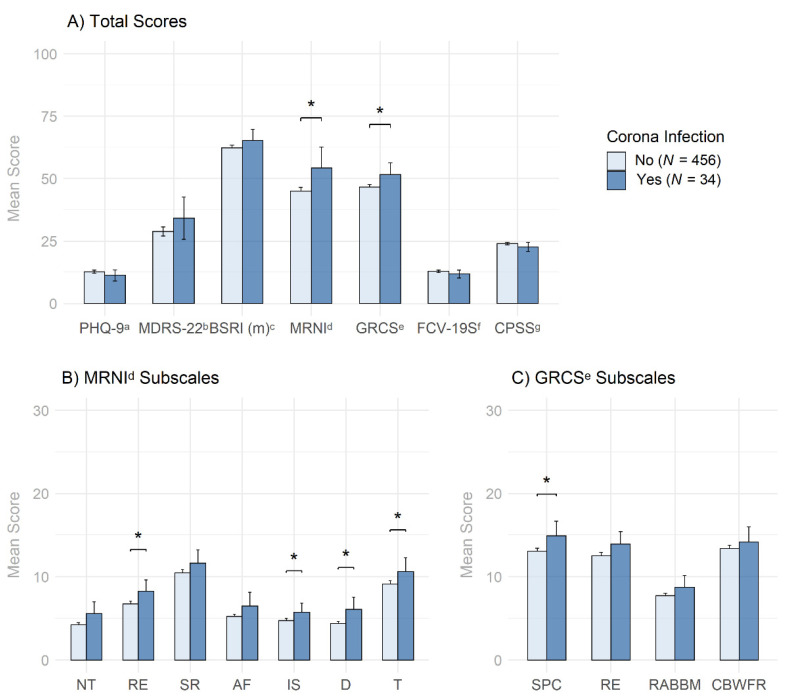
Mean score comparisons between men with and without COVID-19 infection and their two- and one-sided 95% confidence intervals. *Note:*
^a^ PHQ-9 = Patient Health Questionnaire-9; ^b^ MDRS-22 = Male Depression Risk Scale-22; ^c^ BSRI (m) = Bem Sex-Role Inventory (masculinity subscale); ^d^ MRNI = Male Role Norms Inventory (subscales: NT = Negativity toward Sexual Minorities; RE = Restrictive Emotionality; SR = Self-reliance through Mechanical Skills; AF = Avoidance of Femininity; IS = Importance of Sex; D = Dominance; T = Toughness); ^e^ GRCS = Gender Role Conflict Scale (subscales: SPC = Success, Power, Competition; RE = Restrictive Emotionality; RABBM = Restrictive Affectionate Behavior Between Men; CBWFR = Conflicts Between Work and Leisure–Family Relations); ^f^ FCV-19S = Fear of COVID-19 Scale; ^g^ CPSS = COVID-19 Pandemic Stress Scale; * = *p* < 0.05.

**Figure 3 behavsci-12-00080-f003:**
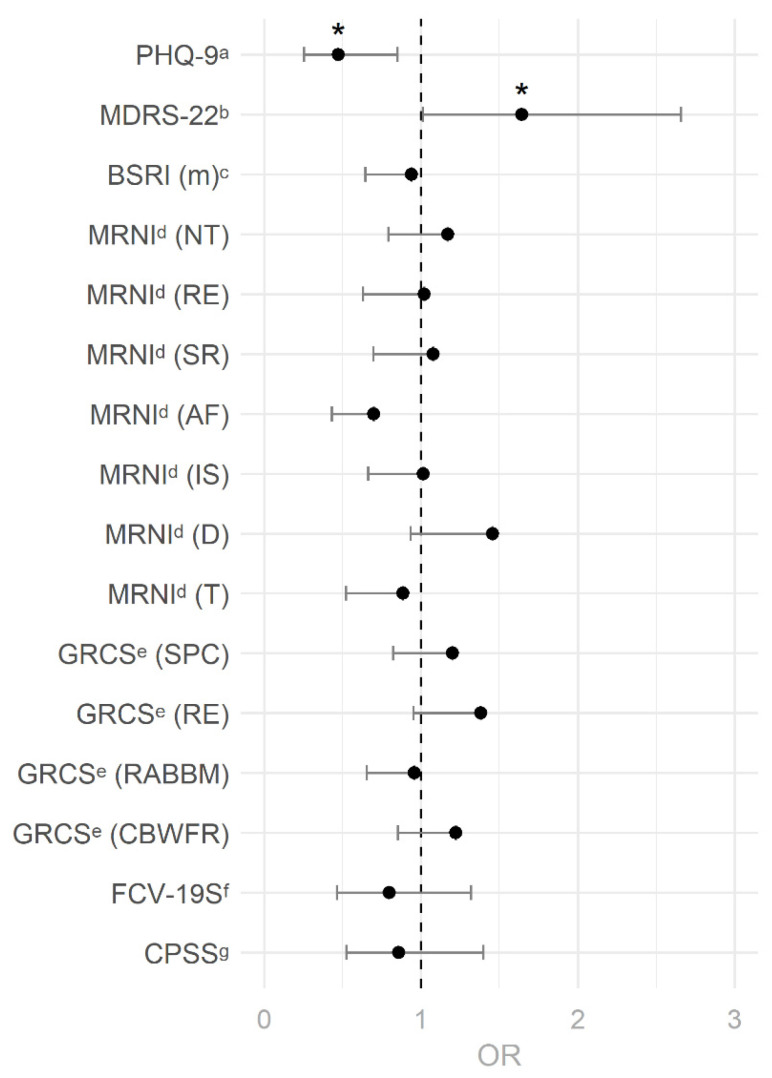
Standardized odds ratios for COVID-19 infection and their two- and one-sided 95% confidence intervals. *Note**:* OR = Odds ratio; ^a^ PHQ-9 = Patient health Questionnaire-9; ^b^ MDRS-22 = Male Depression Risk Scale-22; ^c^ BSRI (m) = Bem Sex-Role Inventory (masculinity subscale); ^d^ MRNI = Male Role Norms Inventory (subscales: NT = Negativity toward Sexual Minorities; RE = Restrictive Emotionality; SR = Self-reliance through Mechanical Skills; AF = Avoidance of Femininity; IS = Importance of Sex; D = Dominance; T = Toughness; ^e^ GRCS = Gender Role Conflict Scale (subscales: SPC = Success, Power, Competition; RE = Restrictive Emotionality; RABBM = Restrictive Affectionate Behavior Between Men; CBWFR = Conflicts Between Work and Leisure–Family Relations); ^f^ FCV-19S = Fear of COVID-19 Scale; ^g^ CPSS = COVID-19 Pandemic Stress Scale. * = *p* < 0.05.

**Figure 4 behavsci-12-00080-f004:**
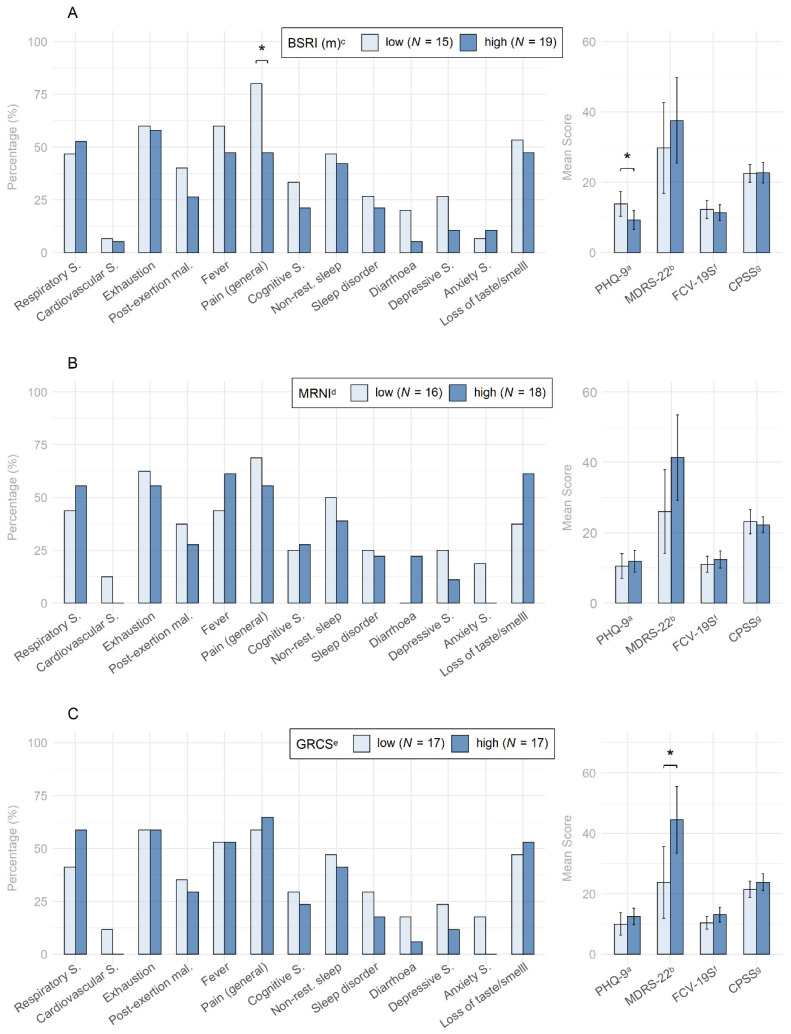
(**A**–**C**) depict differences in COVID-19 and depression symptoms, fear of COVID-19, and COVID-19 pandemic-related stress between men with low and high traditional masculinity. (**A**) compares the frequency of reported symptoms by high versus low masculine role orientation. (**B**) compares the frequency of reported symptoms by high versus low endorsement of traditional masculinity ideologies. (**C**) compares the frequency of reported symptoms by high versus low gender role conflict. *Note: S.* = symptoms, *mal.* = malaise, *low/high* = below/above the median. ^a^ PHQ-9 = Patient health Questionnaire-9; ^b^ MDRS-22 = Male Depression Risk Scale-22; (masculinity subscale); ^d^ MRNI = Male Role Norms Inventory; ^e^ GRCS = Gender Role Conflict Scale; ^f^ FCV-19S = Fear of COVID-19 Scale; ^g^ CPSS = COVID-19 Pandemic Stress Scale. * = *p* < 0.05.

**Table 1 behavsci-12-00080-t001:** Descriptive statistics for the sample.

	Total (*N* = 490)		No CV19 (*N* = 456)		CV19 (*N* = 34)			
	*N* (%)	*M* (SD)	*N* (%)	*M* (SD)	*N* (%)	*M* (SD)	Test-Statistic (*df*)	*p*
**Age**		25.7 (9.8)		25.7 (9.8)		25.9 (9.5)	−0.10 (448)	0.924
**Nationality**							26.63 (6)	**<0.001 *****
Swiss	71 (14.5)		62 (13.6)		9 (26.5)			
German	358 (73.1)		342 (75.0)		16 (47.1)			
Austrian	43 (8.8)		36 (7.9)		7 (20.6)			
Luxembourger	4 (0.8)		4 (0.9)		0 (0)			
Liechtensteiner	1 (0.2)		1 (0.2)		0 (0)			
Belgian	1 (0.2)		0 (0)		1 (2.9)			
Other	12 (2.4)		11 (2.4)		1 (2.9)			
**Sexual orientation**							0.88 (4)	0.927
Heterosexual-identified	361 (73.7)		334 (73.2)		27 (79.4)			
Gay/Lesbian-identified	39 (8.0)		37 (8.1)		2 (5.9)			
Bisexual-identified	67 (13.7)		63 (13.8)		4 (11.8)			
Asexual-identified	5 (1.0)		5 (1.1)		0 (0)			
Other	18 (3.7)		17 (3.7)		1 (2.9)			
**Marital status**							1.45 (2)	0.484
Single	311 (63.5)		291 (63.8)		20 (58.8)			
In a relationship	168 (34.3)		154 (33.8)		14 (41.2)			
Separated after permanent relationship	11 (2.2)		11 (2.4)		0 (0)			
**Education**							3.69 (3)	0.296
None completed	10 (2.0)		10 (2.2)		0 (0)			
Secondary education	348 (71.0)		325 (71.3)		23 (67.6)			
Tertiary education	106 (21.6)		99 (21.7)		7 (20.6)			
Other	26 (5.3)		22 (4.8)		4 (11.8)			
**Yearly household income (in CHF)**							1.88 (2)	0.392
<25,000	233 (47.6)		220 (48.2)		13 (38.2)			
25,000–50,000	92 (18.8)		86 (18.9)		6 (17.6)			
>50,000	165 (33.7)		150 (32.9)		15 (44.1)			
**Due to CV19 Pandemic ^†^**								
Status Loss	72 (14.7)		67 (14.7)		5 (14.7)		0 (1)	1
Financial problems	100 (20.4)		92 (20.2)		8 (23.5)		0.06 (1)	0.804
Job insecurity	112 (22.9)		102 (22.4)		10 (29.4)		0.54 (1)	0.464
Job loss	41 (8.4)		37 (8.1)		4 (11.8)		0.18 (1)	0.674
Registration with the employment center (RAV)	34 (6.9)		30 (6.6)		4 (11.8)		0.64 (1)	0.425
Existential threat	95 (19.4)		86 (18.9)		9 (26.5)		0.74 (1)	0.391
**Psychological Disorder ^†^**	117 (23.9)		112 (24.6)		5 (14.7)		1.19 (1)	0.275
**Psychotherapy ^†^**	95 (19.4)		92 (20.2)		3 (8.8)		1.93 (1)	0.164
**Psychiatric Medication** ^†^	64 (13.1)		62 (13.6)		2 (5.9)		1.05 (1)	0.306
**Depression Cutoff**								
PHQ-9 (≥10)	322 (65.7)		305 (66.9)		17 (50.0)		3.29 (1)	0.070
MDRS-22 (≥51)	67 (13.7)		58 (12.7)		9 (26.5)		3.07 (1)	**0.046 ***
**PHQ-9 ^a^**		12.6 (6.6)		12.7 (6.6)		11.3 (6.3)	1.21 (488)	0.225
**MDRS-22 ^b^**		29.2 (20.6)		28.8 (20.3)		34.1 (24.4)	−1.45 (488)	0.169
**BSRI (m) ^c^**		62.5 (14.6)		62.2 (14.6)		65.4 (15.2)	−1.20 (488)	0.116
**MRNI-SF ^d^**		45.6 (21.4)		44.9 (20.7)		54.4 (28.9)	−1.87 (35.6)	**0.025** *****
**GRCS-SF ^e^**		47.0 (12.6)		46.7 (12.2)		51.7 (16.0)	−1.78 (35.9)	**0.042 ***
**FCV-19S ^f^**		12.8 (4.9)		12.9 (4.90)		11.8 (4.60)	1.28 (488)	0.200
**CPSS ^g^**		23.9 (5.2)		24.0 (5.15)		22.6 (5.38)	1.52 (488)	0.130

*Note*: *N* = number of participants, *M* = mean, *SD* = standard deviation, test-statistic = *t*-value for continuous, *χ*^2^-value for categorical variables, *df* = degrees of freedom, *p* = *p*-value, CV19 = COVID-19. Subjective social status was dichotomized using a median-split. **^†^** Assessed in self-report ^a^ PHQ-9 = Patient Health Questionnaire-9; ^b^ MDRS-22 = Male Depression Risk Scale-22; ^c^ BSRI (m) = Bem Sex-Role Inventory (*m* = masculinity subscale); ^d^ MRNI-SF = Male Role Norms Inventory–Short Form; ^e^ GRCS-SF = Gender Role Conflict Scale–Short Form; ^f^ FCV-19S = Fear of COVID-19 Scale; ^g^ CPSS = COVID-19 Pandemic Stress Scale. Significant results are displayed in bold. * = *p* < 0.05, *** = *p* < 0.001.

**Table 2 behavsci-12-00080-t002:** Correlation matrix for study variables.

	*M*	*SD*	1	2	3	4	5	6
1. PHQ-9 ^a^	12.6	6.6	–					
2. MDRS-22 ^b^	29.2	20.6	**0.64 *****	–				
3. BSRI (m) ^c^	62.5	14.6	**−0.36 *****	−0.10	–			
4. MRNI-SF ^d^	45.6	21.4	−0.06	**0.17 ****	**0.33 *****	–		
5. GRCS-SF ^e^	47.0	12.6	**0.41 *****	**0.48 *****	0.01	**0.39 *****	–	
6. FCV-19S ^f^	12.8	4.9	**0.29 *****	**0.28 *****	**−0.16 ****	**−0.15 ****	**0.18 *****	–
7. CPSS ^g^	23.9	5.2	**0.31 *****	**0.28 *****	0.03	−0.04	**0.24 *****	**0.58 *****

*Note: M* = mean, *SD* = standard deviation. *p*-values were adjusted for multiple testing using the Holm–Bonferroni method. ^a^ PHQ-9 = Patient Health Questionnaire-9; ^b^ MDRS-22 = Male Depression Risk Scale-22; ^c^ BSRI (m) = Bem Sex-Role Inventory (masculinity subscale); ^d^ MRNI–SF = Male Role Norms Inventory–Short Form; ^e^ GRCS–SF = Gender Role Conflict Scale–Short Form; ^f^ FCV-19S = Fear of COVID-19 Scale; ^g^ CPSS = COVID-19 Pandemic Stress Scale; ** = *p* < 0.01, *** = *p* < 0.001. Significant results are displayed in bold.

## Data Availability

A priori defined study hypotheses, statistical analyses, and the study specific data set can be retrieved from OSF (https://osf.io/q4pw3, accessed on 9 March 2022).

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
