# Peer review of "Examining the Role of Traditional Masculinity and Depression in Men’s Risk for Contracting COVID-19"

_behavsci, 2022, doi:10.3390/bs12030080_

Round 1

Reviewer 1 Report

I'm thankful for the opportunity to review your paper. 

The paper sounds very interesting and original. The introduction is informative, well-oriented and focused. The statistical analyses are well-conducted and in some point sophisticated. 

The discussion paragraph has been argued integrating literature highlighting the focal points of your findings

The overall outcome of the paper is positive and improves the actual knowledge.  

Author Response

We would like to thank the reviewer for taking the time to review the manuscript. A response letter is attached responding to all reviewer inquiries.

Reviewer 2 Report

- Appreciated the procedure in "The a priori formulated directed hypothesis are retrievable under OSF [osf-link]): learned a trick in the review process! This provides some benefit (better: precision) down the analytical road "one-sided hypothesis tests were used in all the analyses".
- A comment: Male Role Norms Inventory – Short Form MRNI-SF has 7 scales, and some of them restrictive emotionality, negativity toward sexual minorities, dominance, and toughness) seem to overlap with the content of measures from the Gender Role Conflict Scale sub-scales, restrictive emotionality, and restrictive affectionate behavior between men in particular. Meaning there could be more complex models tested among sub-scales from all the measures, and to that end, maybe an expansion of the Table 2. Correlation Matrix for Study Variables" containing such information at sub-scale level might be appreciated in some appendix (so folks can actually test such models, if interested in this). 

Author Response

We would like to thank the reviewer for taking the time to review the manuscript. A response letter responding to all reviewer inquiries is attached.

Reviewer 3 Report

The main findings described in the paper are interesting, they are not fully supported by the empirical analysis.

"While prototypical depression symptoms were negatively associated with the risk of having contracted COVID-19, male-typical externalizing depression symptoms were positively associated with the risk of having contracted COVID-19. For traditional masculinity, no robust association for an increased risk of contracting COVID-19 could be established, while higher male-typical externalizing depression symptoms were associated with an increased risk of having contracted COVID-19."

What is the focus – depression or masculinity?

State clearly - what was pre-registered

Explain the HB-correction
-     Which variables / hypotheses were included in the calculation?
-     How does this relate to pre-resgistration?

-     Be more parsimonious regarding the masculinity measures – why so many at the same time?

-     What is the rationale for that?

-     Why are they not significant

-     Report MC measures

-    Maybe combine them into one measure

-     Confoundation with gender identity?

o    Why is it included as a control?

Sample is skewed

-    Strange: first is depression not significant, but masculinity – then it is the other way round
-    

-    Please add number of observations to tables and figures

-    Male depsrerssion risk scale: indicator of risky behaviors

  •    Social media advertisements

I am worrid about multicollinearty - please report the variance inflation factors in the supplementary material.

-     What about the the single gender item – why not included?

-    Gender Role Conflict Scale – what is measured by this?

"4.1. Summary of Results The present study examined the interaction between mental health outcomes, traditional masculinity and COVID-19 infection." This sentence is actually not true, because the interaction among these measures are not studied or discussed.

too many measures used in the analysis - is fear of covid really necessary?

Author Response

(The authors gave the same response as above.)

Reviewer 4 Report

The study is well conducted. All the parts of the paper are clear.

Author Response

(The authors gave the same response as above.)

Round 2

Reviewer 3 Report

Please refer to my comments in your letter addressing the first review.

Author Response

We thank Reviewer #3 for the comments and suggestions for improvement. Again we provide a response letter with added inquiries and responses. Com- ments added by Reviewer #3 are shown in GREEN, with responses to them again shown in BLUE with a black box around it.
